# Considerations on the Dynamics of Biofidelic Sensors in the Assessment of Human–Robot Impacts

S. M. B. P. B. Samarathunga [1,2,*], Marcello Valori [3], Rodolfo Faglia [1], Irene Fassi [2] and Giovanni Legnani [1,2]

1    Department of Mechanical and Industrial Engineering, University of Brescia, Via Branze, 38, 25123 Brescia, Italy; rodolfo.faglia@unibs.it (R.F.); giovanni.legnani@unibs.it (G.L.)
2    Institute of Intelligent Industrial Technologies and Systems for Advanced Manufacturing, National Research Council of Italy, Via Alfonso Corti, 12, 20133 Milan, Italy; irene.fassi@stiima.cnr.it
3    Technology Transfer Directorate, Italian National Agency for New Technologies, Energy and Sustainable Economic Development, Via Martiri di Monte Sole, 4, 40129 Bologna, Italy; marcello.valori@enea.it
*    Correspondence: b.samarathungamu@studenti.unibs.it; Tel.: +39-328-493-3294

**Abstract:** Ensuring the safety of physical human–robot interaction (pHRI) is of utmost importance for industries and organisations seeking to incorporate robots into their workspaces. To address this concern, the ISO/TS 15066:2016 outlines hazard analysis and preventive measures for ensuring safety in Human–Robot Collaboration (HRC). To analyse human–robot contact, it is common practice to separately evaluate the "transient" and "quasi-static" contact phases. Accurately measuring transient forces during close human–robot collaboration requires so-called "biofidelic" sensors that closely mimic human tissue properties, featuring adequate bandwidth and balanced damping. The dynamics of physical human–robot interactions using biofidelic measuring devices are being explored in this research. In this paper, one biofidelic sensor is tested to analyse its dynamic characteristics and identify the main factors influencing its performance and its practical applications for testing. To this aim, sensor parameters, such as natural frequency and damping coefficient, are estimated by utilising a custom physical pendulum setup to impact the sensor. Mathematical models developed to characterise the sensor system and pendulum dynamics are also disclosed.

**Keywords:** collaborative robots; biofidelic sensor; ISO/TS 15066; human–robot interaction; human robot physical interaction; robot safety





## 1. Introduction

As robots continue to be integrated into different industries and work settings, it is crucial to prioritise the safety of human workers during these collaborative efforts [1], and this will become increasingly fundamental considering the worker-centric perspective of the industry 5.0 paradigm [2].

The well-established concept of HRC refers to all those industrial scenarios in which workers and robots share a workspace without being separated by fences and barriers. Whenever a robotic application involves physical contact or when it is reasonably foreseeable, it is important to consider the physical human–robot interaction (pHRI) to ensure safety. pHRI issues should be addressed by conducting a risk assessment of the application [3]. To specifically deal with human–robot contact scenarios, presented in the Appendix A, the American standard RIA TR R15.806 [4] proposes guidelines to test force and pressure limits in power and force-limited applications for physical human–robot contact, considering appropriate equipment, setup, procedure, and performance criteria for quasi-static and transient situations. Similar ones will likely be included in the ISO 10218-2 [5] updated version (in preparation: ISO/FDIS 10218-2 Annex N). Recent research has focused on the development of testing procedures for a variety of HRI scenarios, addressing several cases of human–robot contacts [6]. On a general basis, once the possible contact events in the risk assessment are identified, the worst-case scenarios are

established by determining the most critical operating conditions, such as velocity and trajectory. Measurements are then conducted reproducing the contact scenario, substituting the human body part with specific force/pressure measuring devices, accurately placed, fixed, and configured according to the body region characteristics; the obtained values are then compared to the biomechanical pain onset limits reported in ISO/TS 15066 [7]. This measurement procedure is to be repeated for every identified contact incident, starting from the device location and configuration.

In view of these considerations, the implemented force/pressure sensors play a crucial role in ensuring safe interactions between humans and robots. By detecting potential collisions, sensors must meet specific criteria, such as accuracy, sensitivity, fast response times, biomechanical fidelity, durability, and calibration, to provide precise measurements [8–10].

### 1.1. State of the Art

The research community has increasingly addressed the different aspects related to human–robot contact modelling and testing. For example, a semi-ellipsoidal model of the human body model was proposed in [11], and a model has been created for human–robot collisions utilising a mass-spring model [12].

The German Aerospace Center developed lightweight robots with mobility and the ability to interact with humans and uncertain environments. To evaluate their safety during physical interactions, crash tests were conducted to measure the potential risk of injury posed by the robot manipulator. The focus was on unexpected rigid frontal impacts, while excluding injuries caused by sharp edges. Various injury mechanisms and severity indices were evaluated for adaptability to pHRI [13].

Haddadin et al. suggested a safety tree that identifies possible injuries, relevant factors, expected severity, and indicators, taking into account quasi-static and dynamic loads and constrained or unconstrained collisions. They also called for a new low-ranking safety scale [14].

Behrens et al. conducted multiple research projects since 2010, focusing on human–robot collisions [15]. Those studies have delved into pain/pressure threshold limits and first low-level injuries for various body parts, considering separately quasi-static and dynamic (transient) contact, constrained, and unconstrained situations, as well as blunt and semi-sharp surfaces. A human subject study involving 112 participants aimed to determine biomechanical limits for safe human–robot interactions, focusing on pain-onset limits for impacts and pinching contacts. The study carefully addresses the unique characteristics of robots and the safety requirements for human–robot interaction (pHRI) in industrial settings. Moreover, the routine used to deal with signal disturbances is narrowly described, including signal filtering, the appropriate interpolation of pressure acquisitions, and the adjustment of the measured values considering the actual inertia of the pendulum device.

A psychophysiological experiment was conducted to simulate the clamping scenario and to measure the dynamic response of the contact force on the human hand. Specifically, the centre of the palm, thenar eminence, and the forefinger pad were chosen as the soft tissues to be impacted. To quantify the viscoelasticity of these tissues, a nonlinear five-element viscoelastic model was developed through curve fitting of the contact force [16].

Herbster et al. [17] proposed a model to determine contact forces in constrained collisions between humans and collaborative robots. The model allows for iterative determination of the maximum safe velocity, with experimental tests confirming its efficacy. The authors also discuss the experiments involving different cobots and payloads, as well as a comparison of the maximum contact force between the experimental results and the pinching model for different robots. In [18], an experimental study using impact tests to determine the apparent mass of collaborative robots is presented, whereas in [19], a conversion method to ease the validation of free collisions is proposed.

Jenneau et al. [20] proposed a methodology to simulate impact scenarios based on a reduced mass-spring-mass model, used to calculate the parameters of a compliant robot; the

approach enables the prediction of the impact force, taking into account the characteristics of the human body.

In recent years, there has been a notable increase in research on sensors that emulate human-like mechanical sensing, which is used in experimental arrangements to quantify and validate the forces produced when a robot interacts with different parts of the human body. The research conducted by Case et al. [21] introduced a soft pressure sensor designed to measure pressure on deformable body parts without introducing any local stiffness. The sensor, which includes a foam skin and elastomeric layer, deforms alongside the measured body part, ensuring accurate pressure measurement. The sensor's quasi-static performance demonstrates linear capacitance changes with applied force. In [22], pressure pain thresholds in human forearms were measured using various probe areas and compared with a pain-sensing system replicating mechanical nociceptor sensing in human skin and muscle. This system comprises artificial skin, adipose muscle layers, and bone and employs pressure sensors to mimic superficial and deep somatic pain detection mechanisms. The Dynamic Impact Testing and Calibration Instrument (DITCI), described in [23], is used to test "biosimulant" human tissue artefacts. These artefacts mimic the structure of human skin and soft tissue, offering insights into robot–human impact injuries and serving as an alternative to costly cadaver-based testing in industrial applications. Biosimulant skin is crafted from chromed-tanned cowhide leather, while muscle tissue uses gelatine solutions.

Commercial biofidelic sensors are designed to replicate the response of specific human body regions, measuring forces applied by robots and their distribution. These sensors play a pivotal role in evaluating Power and Force Limitation (PFL) in industrial settings, where they can help prevent injuries and mitigate hazards resulting from accidental human–robot contact. PFL risk assessment considers factors such as contact points, robot configurations, and velocities, requiring an understanding of human body thresholds for resisting biomechanical loads. In well-established best practices, deploying biofidelic sensors is indispensable for assessing and validating power and force constraints during impacts. In a study conducted by Zimmermann et al. [8], the safety of collaborative robot applications was validated by comparing three different commercially available biofidelic force-measuring devices. A systematic experimental methodology was developed to compare the devices comprehensively. Dynamic and static loads were applied using a linear motor test machine and a pendulum, while parameters such as compression elements, mass of the moving plate, geometry, and fixation stiffness varied. The results indicated an average peak force difference of 5% between the devices, with differences increasing in softer compression elements and less rigid fixation. The moving plate mass and dynamic behaviour were also found to have an impact on the results.

Scibilia et al. [24] presented an insightful analysis of safety tests conducted on collaborative robots at four European research laboratories. These labs followed standardised procedures to replicate identical collision tests between a sensor and a robot, testing a diverse range of robot motions and impact points. The primary objective was to assess the level of variability observed when conducting the same test under slightly different conditions. The findings revealed that the impact force typically increased linearly with velocity, as suggested by the model proposed in ISO/TS 15066, but various factors, such as the robot controller, sensor fixture, and distance from the robot base, also influenced the results.

Fischer et al. [25] delved into the topic of collaboration between humans and robots in a workspace that lacks physical barriers, with a particular focus on safety. The authors argued for the need for more precise equations to evaluate impact force and pressure during collisions, as current guidelines may not be practical. Through experimentation, the authors identify parameters that should be included in a formal human–robot collision verification model and propose a model to pinpoint the worst-case affected body region in a collision. The study also revealed the importance of additional parameters, such as damping Shore hardness, stiffness, safety settings, and impactor shape, in determining impact force and pressure. The experiments demonstrate a relationship between impactor

geometries, specifically sharp edges versus rounded ones, and the Shore hardness of the impacted body region, which simulates human tissue.

*1.2. Aim of the Paper*

The contemporary landscape of power- and force-limited collaborative robot testing reveals a notable challenge of variability and uncertainty in testing procedures and measurement protocols. This uncertainty pervades users and system integrators, who grapple with the absence of well-defined methodologies for acquiring precise force and pressure data during close human–robot interactions.

Furthermore, the mechanical attributes and dynamic behaviours of biofidelic sensors, encompassing parameters such as moving plate mass, natural frequency, and bandwidth, introduce additional layers of complexity in the quest for accurate data acquisition.

While ISO/TS 15066 provides a foundational reference point, it has notable limitations, including an oversimplified contact model and a less comprehensive treatment of effective mass, spring constants provided for 100 mm$^2$ contact areas without considering their dependence on the shape of the impacting bodies, and difficulty in accurately measuring peak forces in transient contact due to mechanical or electrical filtering of the impact that can significantly change peak values, making comparisons with limit values unreliable [26].

Additionally, accurately capturing peak forces in transient contacts is challenging due to potential mechanical or electrical filtering, which further complicates the comparability of the results with established limit values. Moreover, the existing literature falls short of providing a comprehensive explanation of the viscoelastic properties inherent to specific body regions and their seamless integration into the biofidelic sensor framework and overarching contact model. Addressing these multifaceted limitations is crucial to advancing the field of human–robot interaction testing and achieving greater precision and understanding.

Therefore, this paper seeks to explore physical interactions between humans and robots by conducting tests on a biofidelic measuring device. Specifically, the focus is on the dynamic properties of such a sensor and the factors affecting measuring performance, deriving opportune implications in testing human–robot interactions. This paper presents a comprehensive model of the biofidelic sensor based on theoretical and experimental analyses. With reference to a previous study focused on a comprehensive analysis of commercial biofidelic devices [8], this paper aims to provide further insights, based on the dynamic analyses reported, on the actual limitations of the use of this kind of sensor, suitable also in the development of new devices. Moreover, the mathematical model obtained can be suitable for further investigations on evaluating the performance of similar biofidelic sensors.

## 2. Experimental Campaign on a Commercial Biofidelic Sensor

A biofidelic sensor, illustrated in Figure 1, is designed to mimic human body parts during simulated impacts with the robot. Incorporating compliant materials like elastomers, these sensors are able to deform and conform to irregular surfaces [21]. By adjusting spring and viscoelastic elements, they are able to simulate the mechanical properties of human tissues, such as stiffness and viscoelasticity. The ISO/TS 15066 [7] provides the appropriate spring stiffness parameters corresponding to the human body part.

Two prominent companies, GTE Industrieelektronik GmbH, Viersen, Germany ("GTE Industrieelektronik GmbH" [Online]. Available: https://www.gte.de/ accessed on 14 July 2023) and PILZ, Ostfildern, Germany ("Pilz—Safe automation, automation technology—Pilz INT" [Online]. Available: https://www.pilz.com/en-INT accessed on 14 July 2023), provide human–robot collision measurement equipment to ensure the safe interaction between humans and robots, whose main features are reported in Table 1. Their state-of-the-art testing devices replicate and assess various collaborative scenarios across various industries, including healthcare, manufacturing, and transportation, to enhance the efficiency and security of collaborative robotic systems. The present study delves deeper

into analysing a commercial biofidelic sensor using a suitable physical pendulum setup from theoretical and experimental perspectives.

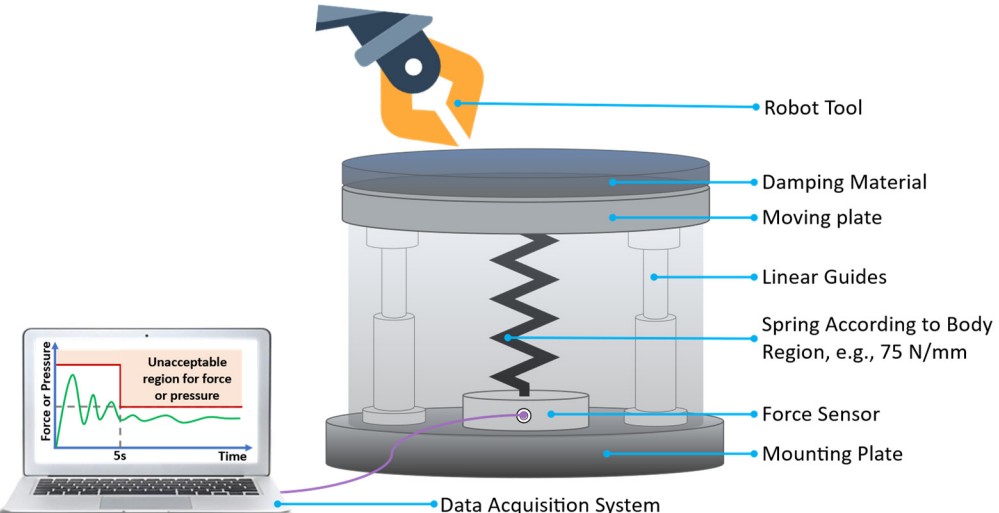

**Figure 1.** Schematic of the biofidelic measuring device.

**Table 1.** Commercially available biofidelic measuring devices.

| Company | Sensor | Available Springs (N/mm) | Force Range (N) | Sampling Rate | Bandwidth |
|---|---|---|---|---|---|
| GTE | COBOSAFE CBSF | 10–150 | 10–300<br>20–500 | ≥ 1 kHz | Not provided |
| | COBOSAFE CBSF—Basic | 75 | 20–500 | ≥ 1 kHz | Not provided |
| PILZ | PRMS | 10–150 | 0–500 | Not provided | Not provided |

### 2.1. Setup and Tests

The physical pendulum (Figure 2) used was a quad-track aluminium bar of mass, $m_b = 0.698$ kg attached to a hinge at a fixed framework, allowing free rotation of the rod. A position adjustable mass $M = 1$ kg and an adjustable position contact point are attached to the rod. The length of the physical pendulum is $L = 0.29$ m, while the variable distance between the hinge and the mobile mass is denoted by $d$, and the separation between the hinge and the contact point is denoted by $l$. The tangential velocity at the contact point is denoted by $v = \omega l$, where $\omega$ is the angular velocity of the rod. The pendulum represents the impacting robot, and the sensor mimics the human body.

Two setups with different configurations were considered, with the sensor placed vertically and horizontally, as shown in Figure 2a,b, respectively. Both included three accelerometers: one attached to the sensor's moving plate to analyse its vibration, another placed near the physical pendulum's contact point, and the third affixed to the base structure to check for vibrations in the structure.

The role of accelerometer sensors was limited to system debugging in this study, and it was not utilised for further dynamic analysis of the biofidelic sensor system. This decision was based on the specific goals of evaluating the sensor's ability to record impact forces and its dynamic properties. Therefore, excluding this data from the subsequent detailed analysis allowed us to focus on the key aspects of the biofidelic sensor's performance.

The pendulum was suitable for simulating lightweight collaborative robots. Section 3 provides a detailed discussion of the analyses and calculations based on the data used to derive the average natural frequency, bandwidth, damping factor, and equivalent mass of the biofidelic sensor. Additionally, the analysis was extended to determine the system's equivalent mass. The equivalent mass of such a pendulum at the contact point depends on the position of the movable mass and the contact point, both adjustable in the setup; the analytical calculation is discussed in detail in the following section. For the test campaign,

it was assumed $l = 0.26$ m, and experiments were repeated for $d$ between 0 m and 0.25 m, generating equivalent masses between 1.67 kg and 2.68 kg (see Section 3.2).

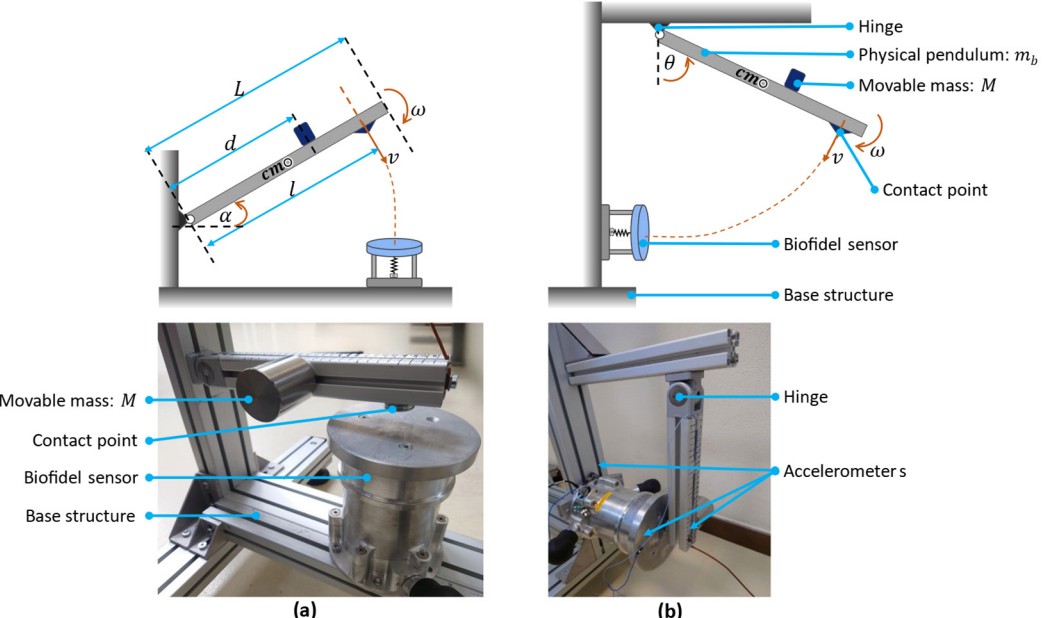

**Figure 2.** Experimental setup: (**a**) Vertical configuration of the biofidelic sensor; (**b**) Horizontal configuration of the biofidelic sensor.

The data collected from the force sensor and accelerometer systems were analysed in MATLAB. Since the different acquisition systems (both proprietary, one dedicated to the biofidelic force sensor, and a National Instrument NI 9233 for the accelerometers), the data were synchronised to be processed. A curve fitting method and fast Fourier transformation were utilised to approximate the force sensor's dynamic behaviour. The natural frequency, bandwidth, and damping coefficient were determined using this method. During these tests, the biofidelic sensor was configured with a spring constant of 75 N/mm and tested both with and without a compression element (damper) rated at a shore hardness of 10 (SH10).

### 2.2. Mathematical Model

This section includes an examination of the dynamic behaviour of the physical pendulum. This investigation resulted in the development of mathematical expressions that govern the equivalent mass of the pendulum, angular velocity, and tangential velocity at the point of contact between the pendulum and the measuring device in various configurations.

#### 2.2.1. Vertical Configuration of the Force Sensor

In this configuration, the biofidelic sensor was set up in a vertical orientation, and a physical pendulum was connected to a vertical bar. When the rod is at rest and an $\alpha$ angle with the horizontal (Figure 2a), the mechanical potential energy $E_p$ can be expressed as follows:

$$E_p = \left( \frac{m_b L}{2} + Md \right) g \sin \alpha \tag{1}$$

with $g$ represents the gravity acceleration. When the rod rotates with an angular velocity $\omega$, the rotational kinetic energy $E_k$ can be written as:

$$E_k = \frac{1}{2} \left( J_0 + Md^2 \right) \omega^2 \tag{2}$$

where $J_0$ is the rotational inertia of the rod about the axis through one end perpendicular to the length: $J_0 = \frac{1}{3}m_b L^2$.

A mass is dynamically equivalent to a pendulum during an impact if it produces the same dynamical effects; this means that they have the same kinetic energy. By indicating by $\overline{M}$ the equivalent mass located in the impact point at a distance $l$ from the hinge, then rotational kinetic energy can be rewritten as in Equation (3):

$$E_k = \frac{1}{2}\overline{M}v^2 = \frac{1}{2}\overline{M}(\omega l)^2 = \frac{1}{2}\overline{M}l^2\omega^2 \tag{3}$$

Hence, Equations (2) and (3) represent the rotational kinetic energy of the same system; by equating them, the equivalent mass of the system is written as:

$$\overline{M} = \frac{J_0 + Md^2}{l^2} = \frac{m_b L^2 + 3Md^2}{3l^2} \tag{4}$$

For a setup with fixed $M$, $m_b$, and $L$, the equivalent mass $\overline{M}$ depends on $d$ and $l$. Therefore, by varying these parameters, different equivalent masses can be obtained.

The system's total energy is conserved if the external forces are negligible. Therefore, by considering Equations (1) and (2), when releasing the pendulum from an initial still condition with an angle $\alpha$, the impact velocity at the contact point can be written as:

$$v = l\omega = l\sqrt{\frac{3(m_b L + 2Md)g\sin\alpha}{m_b L^2 + 3Md^2}} \tag{5}$$

2.2.2. Horizontal Configuration of the Force Sensor

The physical pendulum was affixed to a horizontal bar in this setup; the force sensor was attached to a vertical bar in a horizontal orientation, as illustrated in Figure 2b. Since the same physical pendulum is used, the equivalent mass of the pendulum is given by Equation (4). By denoting the angle $\theta$ between the rod and the vertical direction $(\theta = 0^0$ in the impact configuration), and assuming an arbitrary initial inclination $\theta$ at release, the impact velocity can be written as:

$$v = l\omega = l\sqrt{\frac{3(m_b L + 2Md)(1 - \cos\theta)g}{m_b L^2 + 3Md^2}} \tag{6}$$

*2.3. Dynamic Model of the Biofidelic Measuring Device*

2.3.1. Dynamic Model of the Force Sensing Apparatus

In the first steps of the dynamic study of the sensor, the damping element was not attached to it. Therefore, the force sensor is modelled as a mass-spring-damper system with the following parameters:

- Mass of the moving plate (damping mass), $m$: includes the mechanical guiding parts of the sensor. It influences the system's inertia and response to the impact force.
- Spring constant, $k$: represents the stiffness of the human body region that we aim to replicate with the biofidelic sensor.
- Damping constant, $c$: represents the sensor's viscous behaviour generating internal energy dissipation.
- Displacement, $x$: the displacement of the contact surface typically measured from its rest position.

Figure 3a shows the tested commercial sensor, featuring a moving plate constrained by linear drives, on top of an internal spring, which is designed to mimic the stiffness of a specific human body region. Figure 3b shows the sensor dynamic model, conceptualised as a spring-mass-damper system; the applied force (impact force) is $f_i$, the measured force from the force sensor is $f_o$, and $x$ is the displacement of the contact surface.

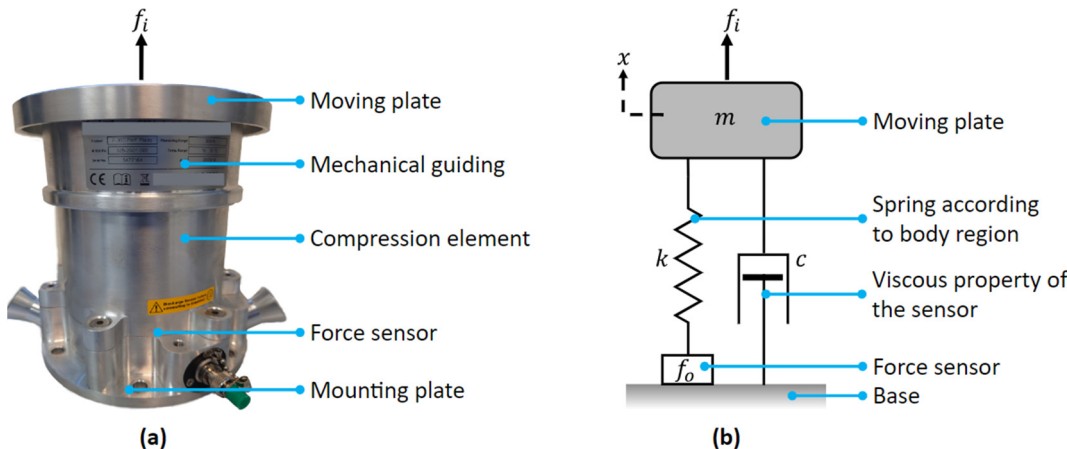

**Figure 3.** (**a**) Commercial biofidelic sensor; (**b**) Dynamic model of the biofidelic sensor (the internal structure is outlined in Figure 1).

By indicating $x$ as the displacement of the moving plate from the rest position, the dynamic equation can be derived by taking into account the dynamic equilibrium of the sensor's mass, as in Equation (7):

$$f_i = m\ddot{x}(t) + c\dot{x}(t) + kx(t) \tag{7}$$

By taking the Laplace transformation, the transfer function of the system can be written as:

$$\frac{X}{F_i} = \frac{1}{ms^2 + cs + k} \tag{8}$$

with $s$ representing the complex frequency variable in the Laplace domain. The frequency response characteristics of the system are obtained by substituting $s$ with $j\omega$ (where $\omega$ is the angular frequency).

Since the force measured by the force sensor is proportional to the elastic force of the spring:

$$f_o = kx(t) \tag{9}$$

The transformation function from $f_i$ to $f_o$ can be written by taking the Laplace transformation of Equations (7) and (9) when the natural frequency $\omega_n = \sqrt{k/m}$ and the damping factor $\xi = c/\sqrt{2km}$:

$$\frac{F_o}{F_i} = \frac{k}{ms^2 + cs + k} = \frac{1}{\frac{s^2}{\omega_n^2} + \frac{2\xi}{\omega_n}s + 1} \tag{10}$$

We conclude that the sensor's transfer function is of the second order and that the system can be under or over-damped. Moreover, its bandwidth is limited by the ratio $k/m$.

### 2.3.2. Dynamic Model of the Damping Material

When the sensor is used in an impact test, an external layer of viscoelastic material, called a damper, is added on the moving plate (Figure 4) to simulate the viscoelastic component of the human body segments. Over time, various models have been devised to represent these properties [27]. The standard linear solid (SLS), also referred to as the three-parameter model, is an employed method for characterising the properties of materials. Linear springs and dashpots combine to represent elastic and viscous components, respectively. Simpler models like the Maxwell and Kelvin–Voigt, sometimes referred to as the Zener Models, are also used [28]. Several models have been developed to simulate the viscoelastic properties of material [29], and the correct choice is crucial to have a realistic representation of reality. In [30], a lumped parameter model of contact adhering to the Kelvin form of SLS is used. In this paper, we adopted a generalised Maxwell model

incorporating two linear elastic components ($k_1$ and $k_2$) and one linear viscous component $c_1$ (Figure 5). The model is validated in Section 3.3.

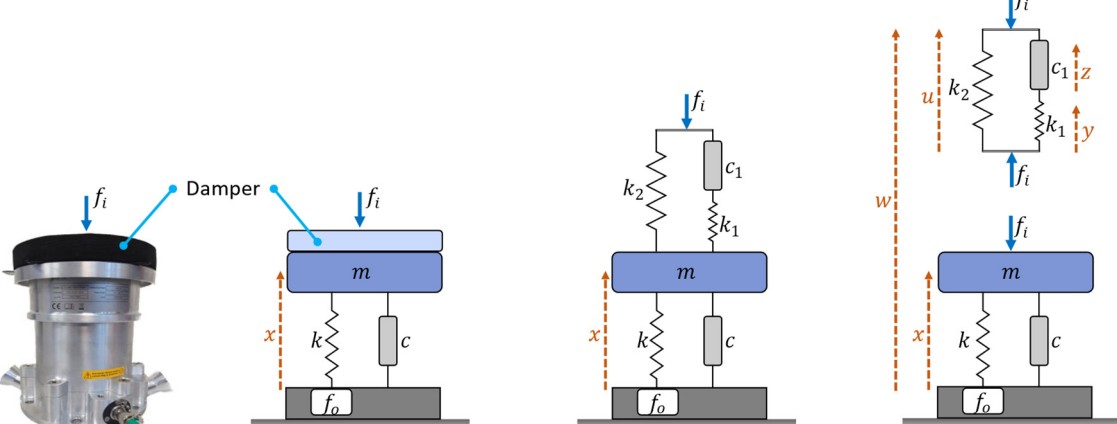

**Figure 4.** The sensor with the damper and successive steps to build its viscoelastic model.

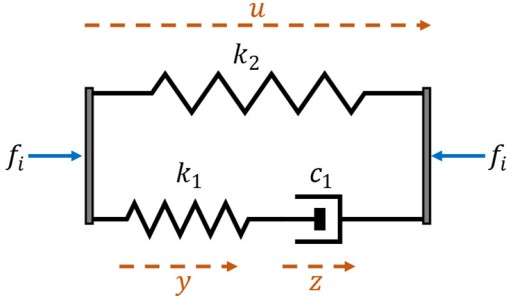

**Figure 5.** The adopted viscoelastic model of the damper.

The model includes two elements in parallel, whose common length is denoted as $u$. Let $f_1$ be the force exerted by the Maxwell model ($k_1$ and $c_1$) and $f_2$ be the force exerted by the parallel spring ($k_2$). The total force $f_i$ acting on the system can be expressed as:

$$f_i = f_1 + f_2 \tag{11}$$

The force $f_1$ is equal to forces in both of the two elements in the series: that of the spring ($f_s$) and the dashpot ($f_d$):

$$f_1 = f_s = f_d = k_1 y = c_1 \dot{z} \tag{12}$$

Since the force exerted by the parallel spring is $f_2 = k_2 u$ and $u = y + z$, the dynamic equation can be written using Equation (11):

$$f_i = c_1 \dot{z} + k_2 u = c_1 \dot{z} + k_2 (y + z) \tag{13}$$

By taking the Laplace transformation, the dynamic equations can be written in the Laplace domain as:

$$\begin{cases} F_i = k_2(Y + Z) + c_1 Z s \\ \\ k_1 Y = c_1 Z s \end{cases} \tag{14}$$

### 2.3.3. Complete Model of the Biofidelic Force Measuring Device

In the development of the biofidelic sensor model, both the sensor and damper models were taken into account, as shown in Figure 4. The damper was considered massless, and its mass was included in that of the moving plate.

Since the damper and the force sensor receive the same force both in the static and dynamic conditions, by using the Laplace transformation of the Equations (7) and (13):

$$F_i = mXs^2 + cXs + kX = k_2(Y + Z) + c_1 Zs \tag{15}$$

According to Figure 4, $w = x + y + z$ and $w = x + u$. The force sensor reading is given by $f_o = kx$. Utilising Laplace transformations enables the variables to be expressed in the Laplace domain:

$$F_o = \frac{kWB}{C} \tag{16}$$

$$F_i = \frac{WAB}{C} \tag{17}$$

$$X = \frac{WB}{C} \tag{18}$$

$$Y = \frac{c_1 WAs}{C} \tag{19}$$

$$Z = \frac{k_1 WA}{C} \tag{20}$$

with:

$$A = ms^2 + cs + k \tag{21}$$

$$B = \{(k_1 + k_2)c_1 s + k_1 k_2\} \tag{22}$$

$$C = mc_1 s^3 + (mk_1 + cc_1)s^2 + \{(k + k_1 + k_2)c_1 + k_1 c\}s + k_1(k + k_2) \tag{23}$$

Finally, it can be obtained:

$$\frac{F_o}{F_i} = \frac{k}{ms^2 + cs + k} = \frac{1}{\frac{s^2}{\omega_n^2} + \frac{2\zeta}{\omega_n}s + 1} \tag{24}$$

This corresponds to Equation (10). We conclude that the transfer function between the contact and the measured force of the sensor does not depend on the damper if the mass of the damper is considered part of the moving plate. However, the elastic and viscous characteristics of the sensor do not influence the bandwidth or the transfer function of the force. On the other hand, the damper has a significant influence on the displacement-to-force transfer function, which is:

$$\frac{F_o}{W} = \frac{kB}{A + B} \tag{25}$$

We highlight that $W$ is the displacement of the contact point (the external layer of the dumper), which is different from that of the moving plate. The displacement of the contact point during impact greatly depends on the viscoelastic characteristics of the damper. It is important that the damper and the spring of the sensor be properly chosen to mimic the viscoelastic properties of the human body segments, whose impact is under analysis.

## 3. Test Results and Discussion

### 3.1. Parameter Estimation of the Force Sensor

The force sensor's parameters, such as the damping constant, natural frequency, and bandwidth, were determined through impact testing. A comprehensive account of the testing process and outcomes can be found in the subsequent section.

### 3.1.1. Analysing the Dynamic Behaviour of the Sensor

The force sensor's dynamic behaviour was analysed using the vertical configuration shown in Figure 2a. The force sensor was tightly fastened to a structure while the accelerometer was positioned on the surface of the moving plate. The first test was conducted by analysing the free oscillation of the moving plate, which was gently struck using a plastic hammer. This process was repeated five times to ensure the accuracy and consistency of the measurements. Typical force and acceleration signals are reported in Figure 6.

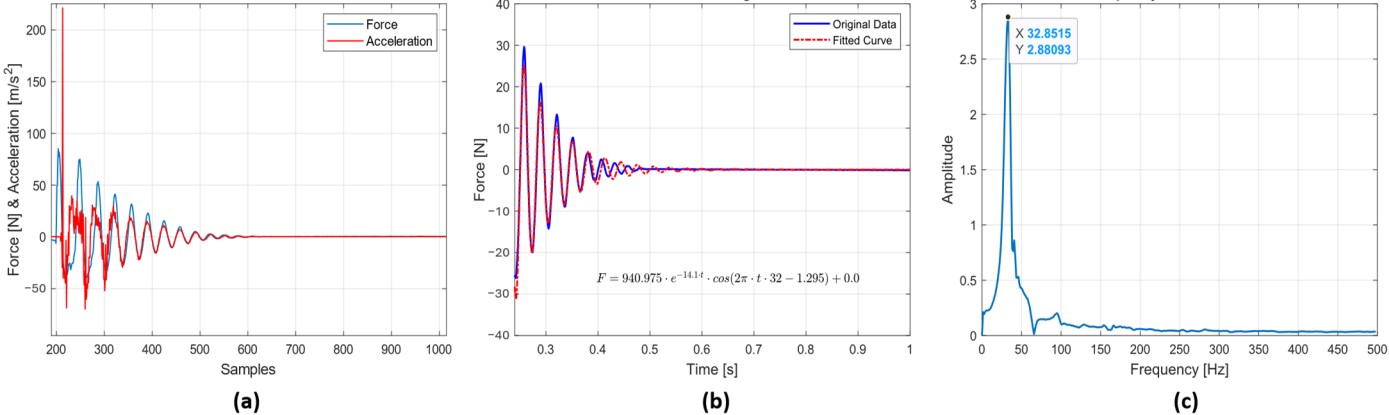

**Figure 6.** Typical signals collected during hammer impact test: (**a**) force and accelerometer readings; (**b**) force fitting; (**c**) Fourier analysis of force signal.

Only the period of free vibration was considered for the analysis. Therefore, the first part of the force sensor data was neglected since it contained the contact period between the hammer and the moving plate. The curve fitting method was utilised to approximate the force sensor data, with the assumption of an underdamped response to the objective function to fit considered as in Equation (26):

$$f(t) = b_1 e^{-b_2 t} \cos\left(\frac{2\pi}{b_3} t + b_4\right) + b_5 \tag{26}$$

in which $b_1$ represents the initial amplitude of the oscillation, serving as a scaling factor for the overall magnitude of the response and it is typically positive. The positive damping coefficient, $b_2$, influences the rate of exponential decay of the oscillations, indicative of the system's damping characteristics. The positive parameter $b_3$ is associated with the frequency of the oscillation and $b_4$ is the phase shift, which can be either positive or negative. Lastly, $b_5$ is a constant term that accounts for any baseline or offset in the force response, and its value can be positive or negative.

The statistical evaluation showed that the objective function was a good fit for the force data. The high R-squared value of 0.953 and low RMSE of 1.14 indicate a high standard of accuracy and quality in the fit.

The damping factor and natural frequency were determined by comparing the objective function to the underdamped response mathematical model in Equation (27).

$$f(t) = A e^{-\xi \omega_n t} \cos(\omega_d t + \phi) + B \tag{27}$$

Where $\omega_n = \sqrt{\dfrac{k}{m}}$, $\xi = \dfrac{c}{2 m \omega_n}$ and $\omega_d = \omega_n \sqrt{1 - \xi^2}$

A total of three tests were performed to derive the variables. By fitting Equation (27) on the experimental data, we obtained the following estimations: The natural angular frequency $\omega_n = 203.2$ rads$^{-1}$ and so the frequency $f_n = \omega_n/(2\pi) = 32.37$ Hz, damped natural frequency $\omega_d = 202.0$ rads$^{-1}$, $f_d = 32.15$ Hz and the damping factor $\xi = 0.0692 \approx 0.07$.

The given results are an average of the three values, and the standard deviation was very little: $\sigma_\% = \frac{\sigma}{\omega_n} = 0.5\%$.

As a low-damped system, it may oscillate if suitably excited by an impact. These oscillations decay more slowly than they would in a predominantly damped system.

However, a Fast Fourier Transformation (FFT) was conducted to confirm this result, revealing the natural frequency to be approximately 32.85 Hz and the damping factor of $\xi \approx 0.07$. These figures are very close to those obtained by the curve fitting methodology, whose results are confirmed.

The mass of the moving plate cannot be measured because it is firmly incorporated in the sensor; however, for the spring, a nominal value of the stiffness is available ($k = 75\,\mathrm{N/mm}$), and it can be removed from the sensor for experimental verification.

An experimental verification of the spring constant was performed using a compression test machine (Universal testing machine, Instron: Model 3366); the experiments were carried out at a rate of 1 mm/min due to the fact that the spring constant for the metal spring exhibited no noteworthy variations with the testing speeds (Figure 7). After conducting compression and cyclic tests, it was concluded that the spring constant value was around 69 N/mm.

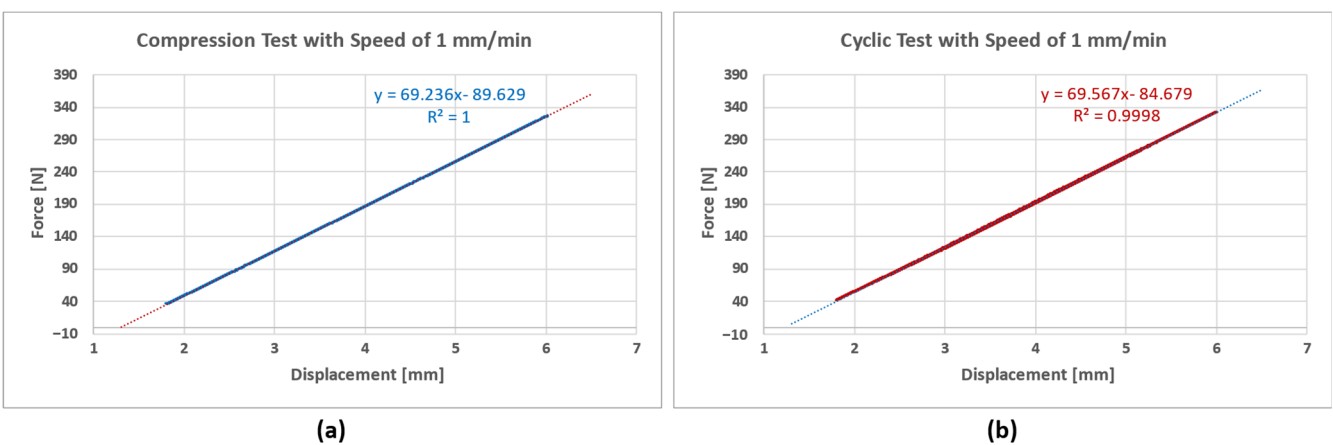

**Figure 7.** Experimental verification of the spring characteristics: (**a**) Compression test; (**b**) Cyclic test.

Determining the combined mass of the sensor's moving plate and linear drives poses a challenge, as the linear drives are fixed to the sensor base, while the moving plate can be easily removed. To overcome this hurdle, an estimation method was employed, utilising the free oscillation of the sensor. The natural frequency of the system, represented by $\omega_n = \sqrt{\frac{k}{m}}$, was assumed to be 203.2 rads$^{-1}$, and the spring constant $k = 69\,\mathrm{N/mm}$, allowing for the approximation of the combined mass of the moving plate and its mechanical guiding, $m$, can be expressed as follows:

$$m = \frac{k}{\omega_n^2} = \frac{69000}{203.2^2} \approx 1.67\,\mathrm{kg}$$

Based on Equation (24), a Bode plot (Figure 8) illustrates the sensor's frequency response. This plot showcases both the magnitude and phase of the system, revealing how different frequency components of the input force are either amplified or attenuated, as well as the phase shift introduced at various frequencies.

The peak at 32.1 Hz in the Bode plot corresponds to the resonant behaviour of the system, which may affect the sensor's stability and performance. For accurate measurement of human–robot contact, the bandwidth of a biofidelic force sensor is critical, as it determines the sensor's ability to respond to force changes at various frequencies. A higher bandwidth sensor can measure motion and vibration at higher frequencies, which is essential for capturing force variations over time during interactions. A low bandwidth

sensor may be unable to track rapid changes in force, leading to errors that can potentially impact the safety and performance of the human–robot interaction system [8].

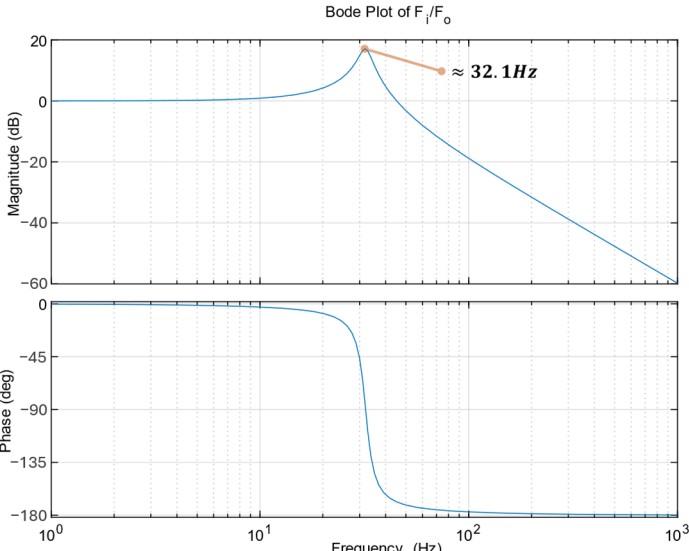

**Figure 8.** Bode diagram of the biofidelic sensor transfer function.

The system of biofidelic sensors is second order. In this context, bandwidth is determined by the frequency $\omega_{BW}$ at which the magnitude response curve decreases by 3 dB from its initial value at zero frequency. Therefore, a two-pole system's bandwidth can be calculated using the following formula [31]:

$$f_{BW} = f_n \sqrt{(1 - 2\xi^2) + \sqrt{(2\xi^2 - 1)^2 + 1}} \tag{28}$$

By assuming $f_n = 32.37$ Hz, and $\xi = 0.07$, the bandwidth of the sensor can be calculated as $f_{BW} \approx 50.12$ Hz.

The sensor exhibited a damping factor of $\xi = 0.07$. Such a property may affect the acquisitions because oscillations are generated during impact, which alter the measure of the force. In relation to this, it is important to point out that just one of the three sensors studied in [8] exhibited oscillations.

Upon further analysis, considering an ideal sensor with the same natural frequency as our model but with a damping factor $\xi = 1$, we identify that the system is critically damped. This implies that the system returns to equilibrium as quickly as possible without oscillatory behaviour. Under these conditions, the bandwidth would be approximately $f_{BW} = 20.83$ Hz. Being the sensor a system of the second order, the approximated time constant of the sensor $\tau$ is $\tau = 2/\omega_n = 2/203$ Hz $\approx 0.01$ s. As highlighted in Figure 9, the sensor cannot reliably measure the force if the impact time is close to or shorter than this period of time. The maximum force is underestimated, and the duration of the peak is elongated. The figures below demonstrate how the sensor behaves when exposed to different impact input forces with varying contact time durations.

The first column of Figure 9 displays the force sensor readings for half-sinusoidal impulses with long contact time $(T_c > \tau)$. The second column shows the response to a double-step impulse, and the third is the response to a half-sinusoidal impulse of a shorter duration $(T_c < \tau)$. The first row considers a measuring sensor having $\omega_n = 203$ rads$^{-1}$ and $\xi = 0.07$, while in the second, it is $\omega_n = 203$ rads$^{-1}$ and $\xi = 1$. It is important to note that the sensor's time constant is $\tau = 0.01$ s. The example figure shows that impulse forces with contact times shorter than 0.01 s cannot be accurately measured by the sensor. When $\xi < 1$ (first row) and the contact force changes rapidly (columns b and c), the sensor oscillates, generating an output that does not represent the real force. When $\xi = 1$ the

sensor has a reaction time approximatively equal to $\tau$, and so filters the variation faster than this value (second row, column c): the measured force is delayed, and the amplitude is highly underestimated. However, the sensor readings are approximately correct when the impulse force has a longer duration, e.g., 0.5 s (first column).

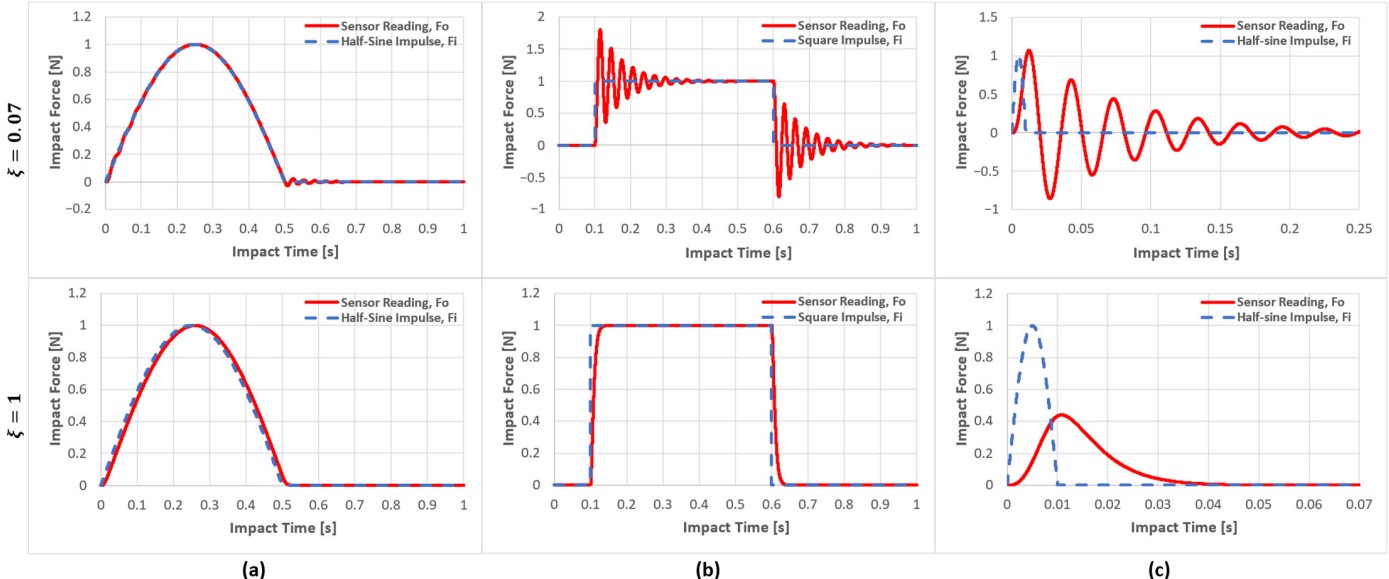

**Figure 9.** Dynamic response of the sensor to impact forces: (**a**) half-sinusoidal impulse with $T_c$ = 0.5 s; (**b**) step impulse with $T_c$ = 0.05 s; (**c**) half-sinusoidal impulse with $T_c$ = 0.01 s. The blue colour represents the real contact force, while the force measured (sensor output) is represented by the red colour.

*3.2. Physical Pendulum*

Experiments were carried out to verify the dynamic behaviour of the sensor during contact with the pendulum, simulating the robot. The vertical configuration was adopted, and the pendulum was released from an initial rest angle $\alpha = 10°$. For the calculations, the distance between the hinge and the contact point ($l$) was set to 0.26 m. Forces and acceleration were collected and analysed using the same procedure illustrated in Section 3.1.1. The test was repeated at different positions of the movable mass ($d$ = 0, 50 mm, 100 mm, 150 mm, 200 mm, and 250 mm). During the impact, the moving plate and the pendulum oscillate as a whole; since the equivalent mass is higher than the case of the isolated sensor, the natural frequency is lower, corresponding to $\omega_n = \sqrt{k/\left(m + \overline{M}\right)}$, with $m$ estimated mass of the moving plate ($m$ = 1.67 kg) and $\overline{M}$ equivalent mass of the pendulum (as per Equation (4)). By adopting an analogue curve fitting method, $\omega_n$ was calculated for all the values, and assessing the angular frequency leads to an estimation of the equivalent mass of the moving plate and the pendulum $m + \overline{M} = \frac{k}{\omega_n^2}$. This value can be compared with that of Equation (4) as a validation of the procedure.

For each configuration, the test was repeated three times, and the standard deviation between the results of the different repetitions was below 1.5%. The average value considered for the comparison is displayed in Figure 10.

Figure 10 shows a strong correlation between the equivalent mass computed using a mathematical model and that computed through the analysis of force sensor data.

Typical results of impact tests using horizontal and vertical configurations are reported in Figure 11 when the sensor was equipped with a damper layer SH10 with a thickness of 24.62 mm and $d$ = 250 mm.

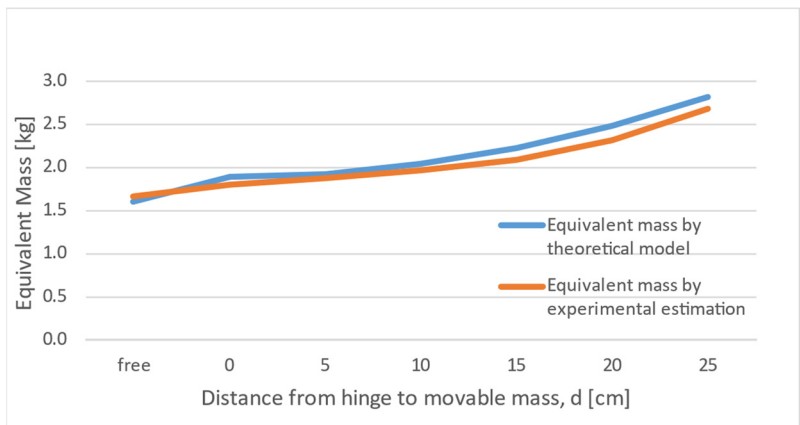

**Figure 10.** Comparison of the equivalent mass found by the theoretical model (Equation (4)) and the experimental estimation.

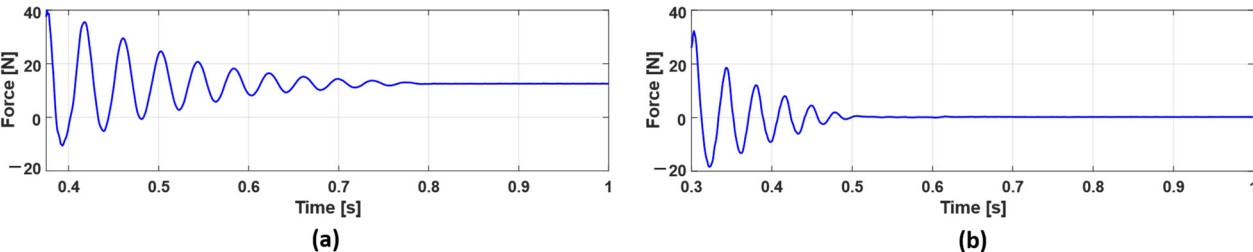

**Figure 11.** Typical example of force Sensor Readings: (**a**) Vertical configuration of the sensor; (**b**) Horizontal configuration of the sensor.

The gross difference between the two cases is the constant force reached at the end of the impact. Trivially, when the horizontal configuration is adopted, the final force is null, while in the vertical case, a positive value is reached as the effect of the pendulum's weight. Moreover, the different dynamic behaviours can be attributed to the friction generated in the linear slides constraining sensor motion, which is definitely more relevant in the horizontal configuration.

It has also been observed that the sensors may indicate a negative force value for some period of time. Since the contact between the pendulum and the sensor is monolateral, the contact force can be only positive or null. However, the oscillation of the moving plate generates this artefact.

Based on the analysis presented, it is worth emphasising that the natural frequency of the sensor is directly impacted by the mass of the moving plate or the combined mass of the moving plate and damping element. Hence, it may be relevant for the moving plate to have a relatively low mass to ensure that the sensor has a high bandwidth, which in turn allows for precise measurement of the impact forces.

### 3.3. Estimating the Parameters of the Damping Material

Experimental tests were conducted to verify the damper model outlined in Section 2.3.2. Specifically, the damper with shore-hardness SH10 and a thickness of 24.62 mm was tested using a compression test machine (Universal testing machine, Instron: Model 3366) equipped with a cylindrical indenter measuring 50 mm in diameter and moving at a velocity of 10 mm/min. The test consisted of five cycles with progressively increasing displacement in each cycle (Figure 12). An indentation speed of 10 mm/min was set to evaluate the properties of the damper material. It is crucial to note, however, that viscoelastic materials can exhibit diverse responses depending on the rate of indentation. Accordingly, additional tests at different speeds are necessary to fully characterise the material's range of responses.

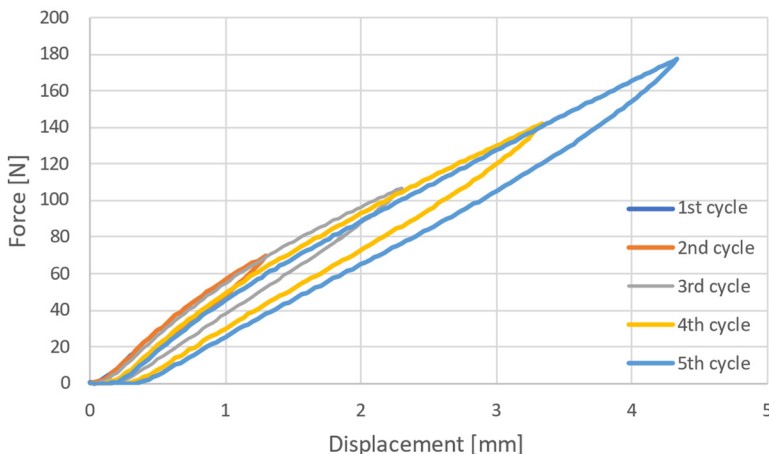

**Figure 12.** Hysteresis of SH10 damping material with a cyclic speed of 10 mm/min.

The protocol uses cyclic ramps at constant velocity, and so considering the model in Figure 5, the force-velocity relation should be as follows:

$$f(t) = c_1\dot{u} + (f_0 - c_1\dot{u})e^{-\frac{t}{\tau}} + k_2 u \text{ and } k_1 = \frac{c_1}{\tau} \tag{29}$$

where $u$ is the deformation, $f_0$ is an integration constant and $c_1$, $k_1$ and $k_2$ are the viscoelastic characteristics of the damper. The value of the constants was estimated by fitting it to the experimental data, and the fitting using a least squares criterion was very good, obtaining $c_1 = 77.9$ Ns/mm, $k_1 = 18.84$ N/mm, and $k_2 = 38.32$ N/mm. For example, in the case of the fifth cycle (maximum deformation about 4 mm, maximum force 175 N), the RMS error was 0.857 N, and the maximum was 6.1 N (see Figure 13 for the case of the fifth cycle). The excellent match between the model and experimental test validates the adopted models and justifies the assumptions of Section 2.3, assuring that the proposed model appropriately represents the whole sensor.

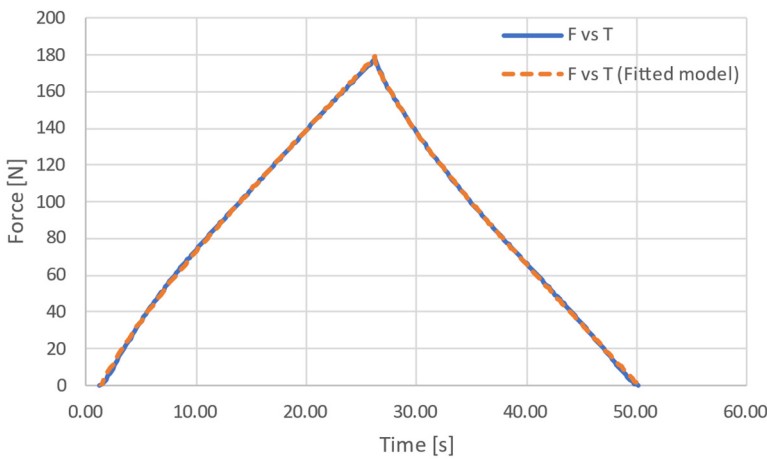

**Figure 13.** Comparison between measured force and force predicted by the model of Equation (29): SH10 damper with a test speed of 10 mm/min of the fifth cycle.

## 4. Conclusions

In conclusion, the effectiveness of biofidelic sensors in physical human–robot interactions hinges on several key factors. First, these sensors must faithfully replicate the mechanical properties of human body segments, encompassing skin, muscles, and bones, to capture interaction forces and ensure safety accurately. This necessitates a better understanding of deformation properties, particularly stiffness and viscosity, with an urgent need for consensus and further research in this area.

Second, an essential requirement for biofidelic sensors is an appropriate bandwidth to measure transient forces, which are critical during sudden or temporary contact between humans and robots. High-frequency transient contacts must be captured for precise measurements. The limitations of existing sensors in certain testing conditions highlight the importance of sufficiently large bandwidth, although guidance on achieving this still needs to be provided in the standards and literature.

This study involved modelling the biofidelic sensor as a mass-spring-damper system and replicating the damping material through an adopted generalised Maxwell model. With the aid of curve-fitting methodology, the natural frequency, bandwidth, and damping factor were determined, and the parameters of the viscoelastic model were estimated. Theoretical and experimental analyses were conducted to verify the sensor's dynamic response when in contact with a pendulum, emulating the behaviour of a robot. It turned out that to improve sensor performance, the mass of the moving plate should be limited to increase the bandwidth, and sufficient damping must be present.

Balancing damping factors and mechanical characteristics is crucial for biofidelity, as it prevents resonance and internal vibrations and ensures dependable data for collaborative robotics applications. Geometric design and calibration processes play significant roles, necessitating improvements in standardisation and documentation. Optimising mechanical parameters, exploring innovative geometries, standardising calibration processes, addressing test conditions, and promoting transparency from manufacturers are key factors to achieve further advancements in the field. Nonetheless, the bandwidth of biofidelic sensors is a critical factor, offering faster response times, adaptability, and resolution of complex forces, but it must be carefully balanced to avoid noise interference. Moreover, various factors, including test curves, load velocities, and sensor fixation, must be considered during collision tests to guarantee consistent and accurate measurements. Measuring errors due to unconsidered monolateral contact must also be avoided.

Incorporating these insights into developing biofidelic sensors can pave the way for safer and more collaborative human–robot interactions, ultimately promoting the increasing implementation of collaborative robots in current practice.

**Author Contributions:** Conceptualisation, methodology, S.M.B.P.B.S., M.V. and G.L.; resources, R.F., I.F. and G.L.; writing-original draft preparation, S.M.B.P.B.S. and M.V.; writing—review and editing, G.L., M.V., I.F., R.F. and S.M.B.P.B.S.; supervision, G.L. All authors have read and agreed to the published version of the manuscript.

**Funding:** This research received no external funding.

**Data Availability Statement:** The study data can be made available upon request to the corresponding author.

**Acknowledgments:** The authors express their gratitude to Giacomo Bianchi at STIIMA-CNR for facilitating access to the accelerometers. In addition, the authors extend their appreciation to Francesco Baldi at the University of Brescia for providing guidance and access to the Universal Testing Machine.

**Conflicts of Interest:** The authors declare no conflict of interest.

## Appendix A. Robot Standards and Human–Robot Impact Modelling

Establishing appropriate safety criteria and regulations is crucial to validating safety. The International Organization for Standardization (ISO) has taken steps towards addressing this challenge through the technical specification ISO/TS 15066: 2016 [7], emphasising hazard analysis to prevent discomfort or injuries during human–robot interactions. However, significant safety concerns still need to be addressed, requiring research and innovation efforts to stay ahead of emerging challenges and ensure that HRC technologies remain safe and effective [32]. International standards, such as ISO 10218-1 and 10218-2, represent the basis for ensuring the safety of industrial robots [33,34]. ISO/TS 15066 particularly focuses on collaborative robot safety. It is important to note that ISO/TS 15066 is a technical specification that is not harmonised by the Machinery Directive. These standards

and specifications (Figure A1) provide guidelines for risk assessments, protective measures, and safety requirements to mitigate potential hazards in human–robot interactions [35]. Some aspects of ISO/TS 15066 are expected to be integrated into the upcoming version of ISO EN 10218-2, which is currently under review [5].

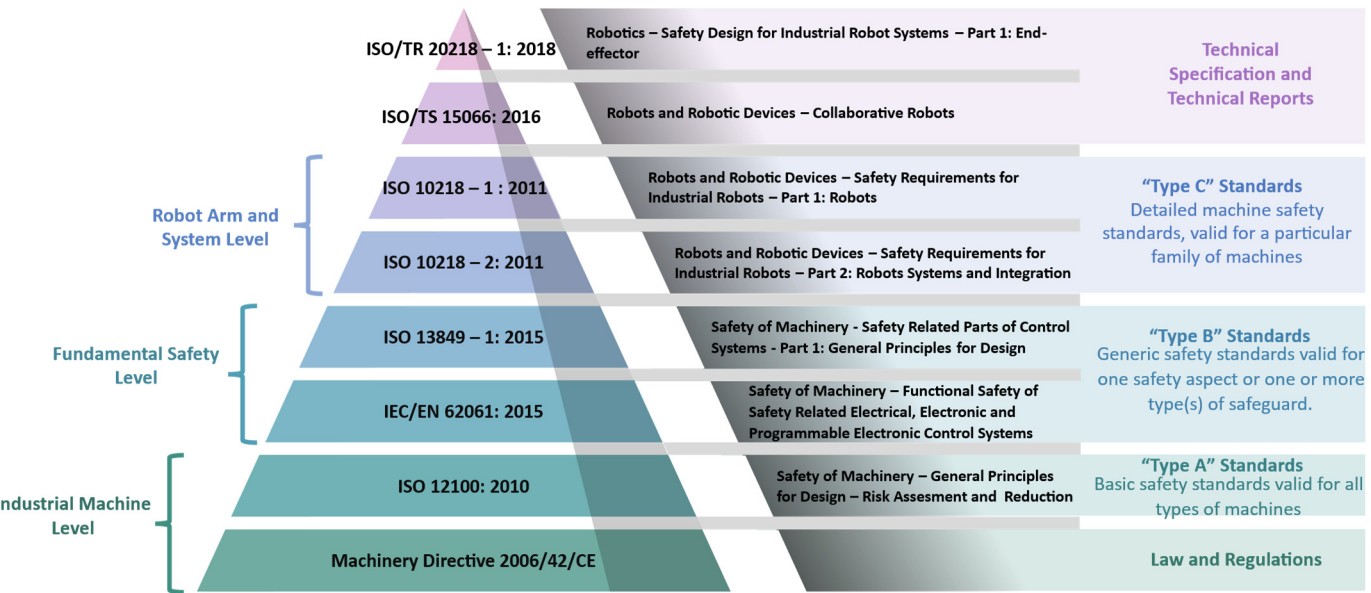

**Figure A1.** International Standards for the Safety of Human–Robot Collaboration.

An impact or contact model is necessary for evaluating robot behaviour in contact situations, helping to develop and validate risk assessment and reduction methods using easily measured signals. The impact model suggested in ISO/TS 15066, which is the one considered in the current practice, assumes a two-body collision with an elastic element in the contact interface, as shown in Figure A2, utilises three key factors: (1) equivalent masses for both the robot and human-robot parts, (2) equivalent stiffness of deformable components and tissues (with a focus on human segment deformation), and (3) collision force is proportional to deformation through stiffness.

In Figure A2a, one object represents the equivalent inertia of the impacted human body region ($m_H$) and the other represents the reflected inertia of the robot manipulator ($m_R$) at the point of contact. The compressibility of the body region acts as the linear elastic element $k$, the contact area $A$, and the relative velocity $v_{rel}$ of the robot, with respect to the body, are the quantities involved. The same contact forces are generated in the model of Figure A2b if the equivalent mass $\mu$ impacts a rigid wall of infinitive mass [7]. Although it is well acknowledged that the equivalent mass of a robot should take into account several factors [36], the Technical Specification prescribes the calculation of the equivalent mass as follows:

$$\mu = \frac{m_H \, m_R}{m_H + m_R} \tag{A1}$$

So, the experimental setting schematically indicated in Figure A2c can be used to experimentally measure the contact forces using a so-called biofidelic sensor (BS).

So, the maximum deformation energy $(E)$ that corresponds to the maximum deformation $\Delta x$ can be calculated as a function of the maximum force or maximum pressure values by using Equation (A2), $(F = k\Delta x)$:

$$E = \frac{F_{max}^2}{2k} = \frac{A^2 P_{max}^2}{2k} \tag{A2}$$

A key assumption of ISO/TS 15066 is that the deformed body region absorbs the entire kinetic energy of relative motion, and so the amplitude of the peak of force can be estimated:

$$E = \frac{F_{max}^2}{2k} = \frac{1}{2}\mu v_{rel}^2$$

$$F_{max} = \sqrt{k\mu}\, v_{rel} \tag{A3}$$

To conduct experimental studies on forces, a single manipulator or an experimental device (impactor) can be used to collide with a biofidelic sensor and after analysing the impact forces at a specific $v_{rel}$, $m$, and $k$, Equation (A3) can be utilised to predict the contact forces for different numerical values of these parameters.

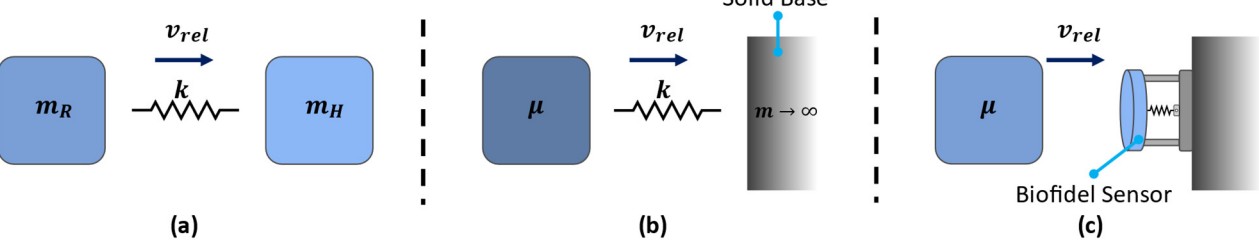

**Figure A2.** (**a**,**b**) Contact model for transient contact suggested in ISO/TS 15066; (**c**) suggested experimental verification using a Biofidelic Sensor.

Human–robot contact analysis typically differentiates between "transient" (initial 0.5 s of impact) and "quasi-static" phases. While the accepted force is greater in the transient phase, the significance of each phase can vary based on specific contact dynamics (Figure A3).

- When a robot impacts a human body part without any constraint, allowing the body part to move freely upon impact, the contact is purely transient, as the human body part can freely move away afterwards.
- Pure quasi-static contact happens when a robot presses a human's body part against a stationary object, like a wall or a table, at low velocity, and the impacting phase can be neglected. These situations can typically occur while doing tasks, like collaborating on object manipulation or passing objects to each other. The force of interaction depends on the deformability of body segments and the time required for the robot to stop when the impact is detected.

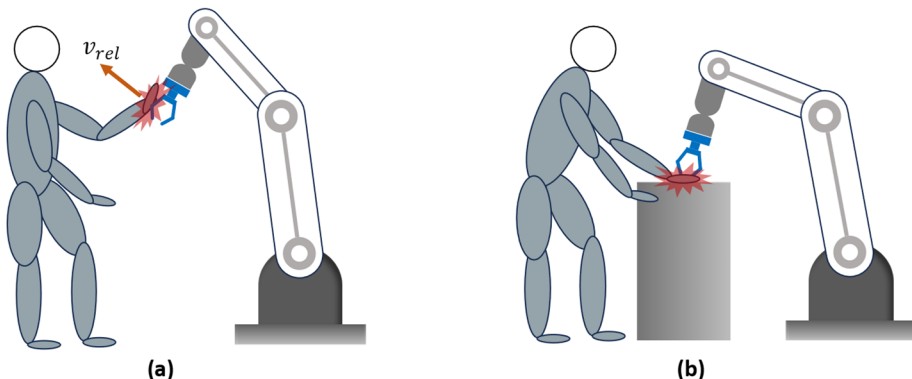

**Figure A3.** Impact Scenarios: (**a**) Transient Impact; (**b**) Quasi-static Impact.

The ISO/TS 15066 incorporates a body model developed based on a pain threshold study conducted at the University of Mainz, Germany. The model has established the highest permissible pressure/force values for different parts of the human body based on the 75th percentile of the study's results.

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
