# Peer review of "Considerations on the Dynamics of Biofidelic Sensors in the Assessment of Human–Robot Impacts"

_machines, doi:10.3390/machines12010026_

Round 1

Reviewer 1 Report

Comments and Suggestions for Authors

In this paper, an introduction about impact safety between robots and humans is given. Referring to this, a mechanical structure is considered which can be used in the analysis of robots impacting the objects in their surrounding. After the discussion of the dynamic model of the structure its identification is conducted using a pendulum structure in two configurations. The results are given in the paper followed by a discussion.

Comments – questions:

    • The term “Fly Wheel” in figure 4 (and later all across the paper) may be confusing. I presume that the intention here was to use an expression like “damping mass”, flywheel assumes rotational motion and might not be the best choice of words here. It may be a good idea to replace the term flywheel with a more suitable one, or at least give a strong motivation for its use.
    • Regarding equation (11): in order to be called Laplace transformation, the variable "s" must be present. If "s" is exchanged with "jw", the result is the frequency characteristic of the system, which is valid during a steady state forced response. Please, be careful about this.
    • Equation (13a): the middle term in the denominator of the third part is missing the variable "s"
    • Equation (13b): likewise, the middle term is missing the variable "s"
    • The two paragraphs starting at line 551 seem to be a bit questionable. With a damping factor of 1, the system is critically damped, which means it is at the boundary of not showing oscillatory behavior. This should be clearly stated here. Additionally, the statement tau=2/omega is problematic. First, the index is missing after the omega. Second, this statement itself should be re-checked – is the factor 2 really necessary?
    • In 3.2 – it should be clearly explained, how this calculation was conducted. In the vertical configuration: after the impact the masses of the flywheel and the pendulum are joined. In the horizontal configuration: after the impact the pendulum and the flywheel are separated – this would explain the higher oscillation frequency of the latter case. Are the previous two sentences true? Please, be precise in the explanation about this. Connected to this: what does the mass m in line 525 refer to? The mass of the flywheel or the joined masses of the pendulum and the flywheel? (since this was the vertical configuration) Please, make these issues absolutely clear, it seems to be a weak point of the paper.
    • Some additional information about the apparatus used for testing the damping material would be welcome in 3.3. It was probably the same machine as the one used for testing the spring in 3.1.1.
    • Can you at least comment on the results from 3.3: can these results be connected to the results from figure 14?

Language remarks:

    • line 102 – "experimentally measure" instead "experimental measure"
    • line 465 – it should be "damper" instead of "dumper"
    • line 467 – "on the other hand" may be a better expression than "on the contrary" here
    • line 518 – use "removed from the sensor" instead of "removed by the sensor"

Author Response

Please find the attached PDF file herewith. Thank you.

Reviewer 2 Report

Comments and Suggestions for Authors  
  1. The article needs to more clearly outline the testing process and techniques, including the use of and performance verification for biorealistic sensors.
  2. The article does not provide enough information to explain how the dynamic characteristics and viscoelastic properties of biorealistic sensors are measured and analyzed.
  3. The article mentions that a sensor was tested, but does not provide sufficient detailed information, such as experimental setup, measurement results, and description of the verification process.
  4. The article does not provide enough data to prove that biorealistic sensors can accurately record impact forces and other key parameters in physical human-machine interactions.
  5. The article does not discuss the limitations and future improvement directions of biorealistic sensors.
  6. The article does not discuss the feasibility and reliability of biorealistic sensors in practical applications.
  7. The article does not provide sufficient evidence to support the effectiveness of biorealistic sensors in physical human-machine interaction testing.
  8. The article does not provide sufficient evidence to prove that biorealistic sensors can be successful in practical applications.
  9. Some experimental results mentioned in the article are insufficient and lack necessary statistical analysis and comparative experiments, which may affect the reliability of the conclusions.
  10. The article does not provide enough new insights or methods to address the problems in physical human-machine interaction testing.
  11. The article does not provide sufficient evidence to prove that biorealistic sensors can be successful in practical applications and may not provide enough information to explain how these sensors are applied in practical testing.
  12. The article should more comprehensively discuss the limitations and improvement directions of biorealistic sensors. A discussion on the accuracy of simulating the mechanical properties of human tissue and biocompatibility is necessary.
  13. A comprehensive contact model and complete mechanical characteristics are not provided, especially a lack of comprehensive explanation for the inherent viscoelastic properties of specific body parts.
  14. Accurately capturing peak forces in transient contact is challenging, and the potential influence of mechanical or electronic filtering further complicates comparing results with established thresholds.
  15. The references are not comprehensive enough, lacking some relevant research and literature, and should be as up-to-date as possible, including the latest research in the field.

Author Response

(The authors gave the same response as above.)

Reviewer 3 Report

Comments and Suggestions for Authors

The objective of this study is to investigate the physical interactions between humans and robots through the implementation of tests and the development of models utilizing biofidelic measuring devices. This is  interesting and meaningful. However, some details in the manuscript still need to be improved.

1. The instruction has reached nearly 9 pages. Therefore, the content in the Instruction section is overly repetitive. It needs to be concise, with a shortened length, and a brief yet comprehensive explanation of the study's hypothesis, purpose, and significance.

2. Line 373- line 423, The authors list a large number of formulas for calculating indicators such as EP, EK, V, fi, etc. How are some formulas obtained? Are these formulas correct?  

3. Line 64-66 When a robot impacts a human body part without any constraint, allowing the body part to move freely upon impact, the contact is purely transient, as the human body part can freely move away afterwards.

As you mentioned in the instruction, when a robot freely impacts a human body part, allowing it to move afterward, the contact is purely transient. In your experiment, regardless of the vertical or horizontal configuration, the sensor remains affixed to the Base structure. How can you use this sensor to simulate the movement of the human body after an impact, i.e. how to simulate transient states with your biosensor?

Author Response

(The authors gave the same response as above.)

Reviewer 4 Report

Comments and Suggestions for Authors

Considerations on the Dynamics of Biofidelic Sensors in the  Assessment of Human-Robot Impacts

In this paper authors have presented hazard analysis and preventive measures for ensuring safety in human robot collaboration. The dynamics of human robot interaction using biofidelic measuring devices are explored. In addition, estimation criteria for natural frequency and damping coefficient is presented. Authors have done a significant work however there are some changes needed in the paper before consideration for the publication.

Authors:

Overall the paper is well written but following suggestion are required to improve it 

1.In Section 2.1, the sentence "The role of accelerometer sensors was limited to system debugging in this study, and the accelerometer data will not be taken into consideration in the analysis presented in the subsequent sections" could be clarified. Specify why accelerometer data won't be considered and its impact on the study

2.Provide a brief explanation for Figure 6, particularly for the subparts (a) and (b). This will help readers understand the representation of the commercial biofidelic sensor and its dynamic mode

3.In the "Dynamic Model of the Biofidelic Measuring Device" section (Section 2.3.1), briefly explain the physical meaning of each variable (𝑚, 𝑐, 𝑘, 𝑥) for clarity

4.In Equation (25), it would be helpful to provide a brief explanation of each parameter (e.g., b1​,b2​,b3​,b4​,b5​) to enhance understanding without referring to Equation (26). Can these constants be negative? Or positive only? Demonstrate the reasoning.

5.Figure 12 needs to be revised. Its resolution is not up to the journal’s standards.

6.Literature review should be expanded further. Highlight and rewrite your contributions of the work? What are major advantages of your method? Add citations from past 5 years.

7.Have you considered disturbances/noises? How to tackle it? Perhaps it can be discussed in the literature review.

8.Use SI units. For example, change cm to m

9.Check equation (13a). ‘s’ is missing with . And should it be  or ? Because . Please check it carefully. If necessary, Revise derivations and simulations accordingly.

10.Equation number is wrong. After (23) why is (13b) there? Revise carefully. Again check it whether it is 2 or square root of 2? If not then 13a and 13b are not same.

Comments on the Quality of English Language

Minor editing of English language required

Author Response

(The authors gave the same response as above.)

Round 2

Reviewer 1 Report

Comments and Suggestions for Authors

I find your added corrections and clarifications appropriate.
Regarding my last remark (from the first review), I must admit that I misinterpreted your explanation. Therefore, this comment does not have a valid point and you should disregard it.

Comments – questions:
    • The term flywheel seems to be abandoned, however, it remained on line 440, you may want to change this as well
    • Section 3.3, in line 570 you refer to Figure 7, you probably meant Figure 5 in the current version.

Author Response

Please find the attached pdf file herewith. Thank you.

Reviewer 2 Report

Comments and Suggestions for Authors

It is better now.

Author Response

(The authors gave the same response as above.)

Round 3

Reviewer 1 Report

Comments and Suggestions for Authors

The paper has been noticeably improved during the review process. A nice experiment is presented, I believe readers will have pleasure reading the paper and find it useful for their work.